# The Effect of Different Cropping Systems on Yield, Quality, Productivity Elements, and Morphological Characters in Wheat (*Triticum aestivum*)

**DOI:** 10.3390/plants12152802

**Published:** 2023-07-28

**Authors:** Ramona Aida Paunescu, Elena Bonciu, Elena Rosculete, Gabriela Paunescu, Catalin Aurelian Rosculete

**Affiliations:** 1Syngenta Agro Romania, 73-81 Bucuresti-Ploiesti Street, 013685 Bucharest, Romania; aida.paunescu@syngenta.com; 2Department of Agricultural and Forestry Technology, Faculty of Agronomy, University of Craiova, 13 A.I. Cuza Street, 200585 Craiova, Romania; catalin.rosculete@edu.ucv.ro; 3Department of Land Measurement, Management, Mechanization, Faculty of Agronomy, University of Craiova, 13 A.I. Cuza Street, 200585 Craiova, Romania; 4SCDA Caracal, University of Craiova, 106 Vasile Alecsandri Street, 235200 Caracal, Romania; gabriela.paunescu@edu.ucv.ro

**Keywords:** wheat, cropping systems, technological management, morphology, quality, production

## Abstract

The aim of this work was to study how certain applied cropping systems (conventional systems differentiated by fertilization level or sowing season and subsistence farming) influence yield, quality, productivity elements, and morphological characters in a collection of Romanian and foreign wheat cultivars. The following indicators were evaluated: productive potential (yield), quality (test weight, protein content, wet gluten content, deformation index, sedimentation index, and gluten index), as well as other elements that determine yield (number of ears/square meter, thousand kernel weight, number of grains/ear, and weight of grains/ear) and plant height. The results show that the cropping systems influenced all the elements studied except the thousand-kernel weight. The only characteristics influenced by higher nitrogen fertilization were test weight, protein content, wet gluten content, deformation index, and gluten index. The superiority of a delayed conventional system was shown by the number of grains/wheat ear and the deformation index. Protein content was differentiated between the conventional and the subsistence system, but especially between the low-input and the conventional system. Nitrogen supply is the most important factor for determining wheat productivity and grain quality.

## 1. Introduction

Conventional and intensive agriculture systems represent an environmental challenge [1]. Conventional crops are characterized by higher yields and profits compared to organic ones [2], but better economic performance is supplemented by negative externalities. Thus, from the environmental perspective, conventional crops cause soil depletion, groundwater pollution, and atmospheric contamination, as well as requiring extensive use of agrochemicals [3]. For crop production, a figure of over 11 million km^2^ of land has been estimated; crop production for human consumption accounts for over 21% of food production emissions, equivalent to approximately 2.8 Gt of CO_2_eq [4]. In the field of wheat production, it was estimated that more than 3.8 m^2^ of land is required to cultivate one kilogram of wheat, while the entire wheat chain generates more than 1.5 kg of CO_2_ per kilogram of product [4].

Although the food system as a whole has enhanced agricultural yields through the adoption of monitoring of crop growth, accurate weather prediction technologies, and novel crop protection methods, to meet demand for food commodities and reduce hunger, such a rapid rise in productivity has had a detrimental effect on the environment [5]. To meet the demand for food globally, crop performance must therefore be increased [6]. Nearly 20% of the calories and protein needed by the world’s population are provided by various wheat products [7].

Wheat (*Triticum aestivum* L.) is one of the three main cereals that can ensure world food security (in addition to rice and maize) [8,9]. About 60% of wheat production is used for food, and the concentration of macro- and microelements in the grain is, therefore, of great importance. In developing countries, it contributes to the edible dry matter and daily net intake of calorie consumption by 28% and 60%, respectively [10,11]. Wheat yields have steadily increased since the onset of the Green Revolution, and as a result, there is an increasing need to efficiently supply P to plants while minimizing negative impacts on the environment [12]. However, in some regions, including Romania, it is difficult to convince the farming community to apply full-recommended fertilizer doses for wheat, but it seems possible to improve fertilizer use efficiency by adopting appropriate time and methods of application [13]. For this reason, cost-effective, environmentally friendly, and more productive farming technologies must be developed [13,14,15].

The differences in yields between regions of the world are due to bio-physical factors such as temperature, rainfall amount, and their distribution during the growing season [16,17]. Modern wheat cultivars have achieved a high yield potential in recent decades. However, their sensitivity to environmental factors has increased, and this negative trend is especially visible in Europe [18].

Agricultural ecosystems are characterized by nonlinearity and interactions among multiple factors, challenging the identification of the true role of each factor [19]. Wheat should be cultivated in a way that ensures a high grain yield of adequate quality to meet the requirements of food processing [20]. Genotype, environmental conditions, technological management, and their interaction determine wheat yield, biochemical composition, and quality [21,22]. For example, the results of a bifactorial experiment with 5 genotypes × 4 technologies—farmer practice, high input, high input without nitrogen, and high input without fungicide showed that high-input technology increased yield by 859 kg/ha compared to farmer practice [22]. Farmer practice is a collection of principles that involve lower inputs. On the other hand, high-input agriculture involves the use of chemical fertilizers and pesticides, complex machinery, and fossil fuels, along with a significant investment of financial capital, to grow crops. The results reported by Raj et al. (2023) showed that high-input technology intensified with fungicide improved grain properties regarding milling characteristics, while that intensified with higher doses of nitrogen improved baking characteristics [22]. Other authors have also stated that wheat yield and quality are determined by the combined effect of the environment and agro-technical measures, as well as species and cultivar or adequate management of manure [23,24,25]. Thus, adequately managed manure is a valuable fertilizer that provides several essential nutrients for crops, as well as organic matter, and can thus alleviate the declining organic matter content in agricultural soils [26].

Some results show that the wheat grains from conventional farming systems had a 6 percent higher protein content than wheat from organic farming systems because of the use of mineral fertilizers [27]. There was no significant difference between organic wheat and conventional wheat in macro- and microelement contents. The quality of baked products obtained from conventionally and organically grown wheat was equally good [27]. On the other hand, the technological quality of wheat from organic farming differs in many aspects from the technological quality of wheat from conventional farming. The most significant differences are in the content of crude protein in the grain dry matter and in the parameters that characterize the wheat protein complex quality [28].

However, many results indicated that the response of cultivars to the applied growing conditions varies and showed that individual wheat cultivars respond differently to the agro-technology used in cultivation, so it is important to select the production system individually according to the requirements, genetics, and production capacity of the cultivar [29,30,31,32,33,34].

In this context, the aim of the paper was to study how the applied cropping systems influence yield, productivity elements, quality, and morphological characters in some Romanian and foreign wheat cultivars.

The motivation for this study was the fact that subsistence farming is massively practiced in Romania, and the results obtained could, through their applicative side, convince farmers to use improved technologies that would bring them production and quality gains. Second, we used several statistical methods to show that differences between these technologies can be reproduced using different methods adapted to the targeted purpose (multiple comparison and comparison between two datasets).

The originality of this study lies in the choice of varieties, soil type, and climatic conditions of the experimental area. In this respect, it should be pointed out that in the tested area, climatic conditions frequently change the sowing date, delaying it until early November.

Given that 25 wheat cultivars of different origins, including the most widely grown in the area, were tested over three years, the results obtained give scientific accuracy to this study.

## 2. Results and Discussion

### 2.1. Number of Wheat Ears/SQM

Comparison of the results based on the one-way ANOVA test, averaged over 3 years, showed that there were significant differences between the systems analyzed (Table 1). These differences did not come from comparing low-input conventional and conventional systems but came from comparing conventional with delayed conventional. For conventional, the number of wheat ears/sqm was higher (743 wheat ears/sqm) but not large enough for this difference to be statistically significant.

The delay in sowing resulted in a very significant decrease in the number of wheat ears/sqm (−241 wheat ears/sqm). Significant differences occurred when comparing low-input conventional with subsistence systems and conventional with subsistence systems, as the latter showed a much lower number of wheat ears/sqm compared to the control systems (Table 1).

Delaying sowing can even lead to spring emergence and, thus, reduce the number of plants emerged/cm, which then form far fewer spikes, the decrease being statistically assured. The results obtained by Slafer et al. [35] highlight the influence of technological, genetic, and environmental factors on production and its components, including the number of spikes/sqm and the number of grains per spike. On the other hand, subsistence farming in which the applied technology is very limited (lack of treatments and lack of fertilization) leads to a decrease in spikes/sqm, which is also suggested by other studies, both for wheat [36] and rapeseed [37].

### 2.2. Plant Height

For the Caracal experiment, height was highly differentiated, both between and within systems. No differences were found on the Caracal chernozem between conventional fertilization-differentiated systems and neither between those differentiated by sowing time. Differences of +0.4 cm and −3.4 cm, respectively, were not statistically assured (Table 2). The reductions in height of 7.8 cm between low-input conventional and subsistence, relatively close systems, by reduced fertilization (Table 2) and of 8.2 cm between conventional and subsistence, totally differentiated systems, were significant. The Mann–Whitney test showed higher values of the U indicator relative to the critical U (124.5, respectively 147.5 in relation to 211).

The literature recognizes that height is a trait strongly influenced by both environmental and technological conditions [38,39,40]. The coefficient of variability of height, more than that of yield, is a good criterion to show its effect on yield variability and stability [41].

### 2.3. Number of Grains/Wheat Ear

The number of grains/wheat ear is one of the productivity elements that was influenced by the cropping systems applied (Table 3). The one-way ANOVA test showed that, between systems, the differences between the recorded values were significant, with the calculated F being higher than the critical F.

Taken two by two, the systems in the first comparison (low-input conventional vs. conventional) did not show significant differences, with the means of the number of grains/wheat ear even being identical (40 grains/wheat ear). The second comparison (conventional vs. delayed conventional), showed a significant difference of +4 grains/wheat ear, which supports the one-way ANOVA test. Therefore, this comparison is also the only one that showed statistically assured differences (Table 3).

The Mann–Whitney U test used to compare two data strings, namely low-input conventional and subsistence on the one hand and conventional and subsistence on the other hand, revealed that there were no significant differences between them, as the U values of 222 and 228.5, respectively, were higher than the critical U value (211) (Table 3).

Studies have shown that the number of grains/wheat ear is influenced by the timing of anthesis [42]. In this respect, the pericarp cell elongation during post-anthesis may be responsible for differences in final grain weight between wheat with different carpel weight at anthesis [42].

### 2.4. Weight of Grains/Ear

Weight of grains/ear, another productivity element that was analyzed, was also influenced by cropping systems, although the calculated F value was the lowest and very close to the critical F value (4.90 vs. 3.239) of all F values calculated for the whole experiment (Table 4).

When comparing low-input conventional and conventional systems, the rather small difference was not statistically significant (1.64 g/wheat ear vs. 1.60 g/wheat ear). For the next comparison, also based on the two-factor subdivided plot (PS2F) program, the difference of −0.04 was not statistically assured either. On the other hand, between conventional low-input (1.64 g/wheat ear) and subsistence (1.54 g/wheat ear), the differences were significant through the Mann–Whitney test, where the calculated U value (156.5) was lower than the critical U (211). The same test did not show a significant difference between conventional and subsistence, with the calculated U value (223.5) being lower than the critical U (211) (Table 4).

In agreement with our results, there are other studies that show the fact that the wheat grain yield and quality are influenced by many factors, including genotype (cultivar) [33], habitat conditions (soil and climate) [41], and agricultural practices [31].

The effect of nitrogen (N) on the grain weight of cereals is complex. N plays a key role in crop productivity and significantly affects the grain weight of cereals [43]. The weight of grains per ear is influenced by the number of ears/m^2^ and by the certain limiting factors, such as deficiencies in nitrogen nutrition, the water stress, the excessive temperatures, the temperature differences between day and night during the grain formation and filling period, the foliar and ear diseases, and the attacks of certain pests [44].

The results reported by Ghitau et al. highlight that the variants treated with BCO-2K biostimulator and fertilized with N160P90K90 and N60P60K60 have obtained the highest weight of grain in the ear [45].

Nitrogen (N), a key limiting element for the growth of most crops, plays an important role in maximizing crop yields on a global scale, leading to a significant increase in the consumption of N fertilizers [46,47,48]. Global N fertilizer consumption has increased ninefold over the past 40 years, while grain yield has only increased by 164% [49].

### 2.5. Grain Yield

Our study showed that the cropping system influenced production (one-way ANOVA test), but among the conventional system variants, on the Caracal chernozem, on average over 3 years, only the delayed conventional showed a distinctly significant yield decrease of 16.4 q/ha (data processing by PS2F) (Table 5).

Similarly, Munaro et al. (2020) showed that technological management influences production [50]. Studies by Shi et al. in 2021 showed that the combined effect of management practices and new crops adoption increased yield by 41.8% [51].

The difference of 6.92 q/ha between low-input conventional and conventional was at the borderline of the limiting difference of 7 q/ha, which would classify it as significant. An important role in placing it close to the limit, although normally the difference between the two variants would have been statistically significant, was largely due to the soil on which the experiments were located, namely chernozem—a soil rich in organic matter, extremely favorable for growing wheat.

The U-values calculated by the Mann–Whitney test had the lowest values in the whole experiment: 11 for the comparison between the low-input conventional and the subsistence system; 10 for the comparison between the conventional and the subsistence system, both being well below the critical U-value (211). Therefore, the 3-year cycle of experimentation and the results obtained allow us to recommend that, for the Caracal chernozome, we apply the conventional system technology, even the one differentiated in terms of fertilization but not the one differentiated by delay of sowing. The yield losses of the conventional system are quite high (20–30 q/ha), but its nonrecommendation can only be made after an economic efficiency analysis based on gross and net margins.

Depending on the applied technological management, the cultivars that stood out were Foxx, Rubisko, Armora, and Abund in conventional (average yield over 10,000 kg/ha), Biharia, Abund, Foxx, Bogdana, and Anapurna at conventional low-input (average yield over 9000 kg/ha), Biharia, Ingenio, Avenue, Dacic, and Fox in conventional delayed (average yield over 8000 kg/ha), and Abund, Anapurna, Biharia, Avenue, Combin, Adelina, and Bogdana in subsistence conditions (average yield over 6500 kg/ha) (Figure 1).

Without exception, all tested varieties recorded very significant decreases in yield under subsistence conditions compared to conventional, on average over the 3 years (LD > 0.01% = 790 kg/ha). With two exceptions (Bezostaia and Ingenio), the yields of the tested cultivars were very significantly lower when sowing was delayed compared to sowing in the normal season (LD > 0.01% = 650 kg/ha).

On the other hand, the differentiation of fertilization at the two graduations of the conventional system (conventional and conventional low input), several of the cultivars tested showed yields at the same level, without statistical assurance (LD > 5% = 450 kg/ha): Miranda, Izvor, Pitar, Voinic, Bogdana, Dacic, Biharia, Anapurna, and Combin. Under conditions of reduced nitrogen fertilization, the Foxx and Kapitol cultivars showed very significant yield reductions of over 2000 kg/ha (LD > 0.01% = 780 kg/Ha).

The coefficients of variability (CV) calculated to study the stability of the yield according to the applied technological management revealed an average stability (CV from 10 to 20%) in most of the varieties tested but also instability, especially in foreign varieties (CV over 20%): Rubisko, Foxx, Kapitol, and the Romanian cultivar Armura (Figure 2).

The most stable cultivars in terms of yield obtained according to the year of testing were Izvor and Combin (CV below 5%). The foreign varieties, with one exception, were the most unstable, being strongly affected by the climatic conditions of the test years (CV above 25%). They were joined by the Romanian varieties Pitar, Abund, Bogdana, and Biharia (Table 6).

The most important characteristic studied, namely production capacity, is crucial in making a decision on the use of the appropriate cropping system at a farm, in addition to pedo-climatic, technological, energetical, and financial resources. However, the intensification of production is costly, and long-term use leads to the degradation of the natural environment [15]. An alternative to intensive production may be integrated technology, in which the use of crop protection products is limited to the necessary minimum, and the doses of mineral fertilizers are selected based on the results of soil tests [24,33].

### 2.6. Thousand-Kernel Weight (TKW)

Calculation of the data series from the four technological variants revealed that the thousand-kernel weight is not influenced by the cropping system (Table 7). This is confirmed by subsequent comparisons. In the same vein, the results of Sulek et al. [24] showed that the increase in grain yield resulted from the increase in the number of grains produced by the plant, but it was not correlated with the thousand-kernel weight.

Differential fertilization did not influence the thousand-kernel weight or the delay in sowing. Comparison of data between low-input conventional and subsistence, as well as between conventional and subsistence, through the Mann–Whitney U test did not reveal significant differences, with calculated U values (214 and 300, respectively) being higher than critical U values (211).

In general, the thousand-kernel weight does not differ significantly between different farming systems. At the same time, it was observed that the method of analysis of this experiment is the right one, since the one-way ANOVA test showed that there were no differences between systems, the additional pairwise comparative studies between systems carried out by the PS2F program and Mann–Whitney U confirming the previous result (Table 7).

### 2.7. Test Weight (TW)

Like most of the elements analyzed, test weight also shows significant differences between the cropping systems analyzed (Table 8). Thus, in the pedo-climatic conditions of Caracal, the differences came from the distinctly significant decrease in TW when the nitrogen dose is increased in the conventional system compared to the low-input conventional system and from the very significant decrease in the same indicator when sowing is delayed compared to the conventional system. Differences also arise from comparing the data series where the value of the test weight was higher in the low-dose version of the conventional system compared to the subsistence system and from the higher value in the conventional system compared to the subsistence system (Table 8).

There are many results suggesting the role of technological management in maintaining or improving wheat quality [52,53,54,55].

### 2.8. Protein Content

In terms of protein content, the amount of variance between systems is not equal to or close to the amount of variance within the system. This finding makes the calculated value of F higher than the critical F. Therefore, the differences for protein content are significant (Table 9).

These differences are primarily the result of differential fertilization under the conventional system (low-input conventional and conventional). When a higher dose of nitrogen is applied, the protein content increased very significantly by +2.2%, and when sowing is delayed, the grain accumulates a high protein content, but it is at the same level as that observed at normal sowing, or more correctly, at sowing 2 weeks earlier. Current climatic conditions have made the notion of normal relative.

The Mann–Whitney U test showed that only in the conventional and subsistence systems were there significant differences (13.0% vs. 10.0%) (Table 9).

Rozbicki et al. (2015) have demonstrated that protein can be improved by differentiated technological practices, especially by the level of nitrogen fertilization [56]. This was also evidenced by our studies on a 3-year average on the Caracal chernozome.

Although earlier studies showed that an increase in yield decreases protein content, recent studies have suggested that it is possible to simultaneously increase yield and protein content through proper nitrogen management [57]. The combination of high nitrogen doses and fungicides can maintain protein content despite yield improvement [58]. The grain protein concentration is influenced by the level of yield [59,60]. The inverse relationship between grain yield and protein concentration may prevent breeders from improving these two traits simultaneously [61]. Therefore, efforts to overcome this inverse relationship must concentrate on improving grain protein accumulation per square meter and per grain [62].

### 2.9. Wet Gluten Content

The wet gluten content is influenced by the technological management, with the calculated F value being higher than the critical F value (Table 10).

These differences come both from conventional differentiated input systems where the wet gluten content is very significant (+6.4%) when the nitrogen dose is higher and from differences caused by the time of sowing when the wet gluten content showed a distinctly significant decrease of −4.7% when there was a two-week delay in the sowing date. Significant differences were also observed for the conventional vs. subsistence comparison, as revealed by the Mann–Whitney U test (Table 10).

There are other authors who reported that the wet gluten content values were significantly influenced by genotype, farming system, and their interaction [20,56,63]. The significantly highest wet gluten content was characteristic of the wheat grain obtained from the conventional farming system [20]. On the other hand, too-high nitrogen doses generally lead to a reduction in gluten quality. This is due to an increase in the proportion of the low-particle fraction of gliadin in the protein [20].

### 2.10. Zeleny Index

For this particular characteristic, the interpretation of the results is a little forced and without substance, given that the values recorded were, on average, not below 60 mL, the limit at which flour is considered good for baking (Table 11).

Although the one-way ANOVA test shows that there are significant differences between systems, which are found for the conventional vs. delayed conventional comparison, the truth is that, regardless of the cropping system, wheat for a good baking quality can be obtained (Table 11).

The Zeleny index depends on both gluten quantity and quality; consequently, it has a strong correlation with baking quality [64].

The results reported by some authors suggest that the protein content, wet gluten content, and Zeleny index value were greatly influenced by environment, although they were also significantly influenced by genotype [56,65].

### 2.11. Deformation Index

The deformation index, another important characteristic of bread quality, showed significant differences from one cropping system to another, with the calculated F value being quite high (136.544 vs. 3.239) (Table 12). Although the differences come from the differentiated nitrogen fertilization (−3 mm in conventional) and from delayed sowing (+4 mm in late conventional), we can say that the deformation index is not affected because its values are in the range of 3 to 9 mm, a range that ensures the harmonious development of a dough.

Significant differences were also found for the comparison of two sets of data involving the subsistence system, where the values of the deformation index no longer ensure the quality standard for bakery products. Other authors show that for bakery, the optimum values of the deformation index are in the range of 5 to 13 mm. A deformation index less than 5 mm indicates good but strong or short, which can be slightly improved. A deformation index greater than 20 mm indicates a weak, filamentous gluten, which signals a degradation caused by an attack of wheat bug [66].

### 2.12. Gluten Index

The gluten index is the only quality indicator that is calculated from the wet gluten and deformation index values (Table 13). It is, therefore, also influenced by the cultivation system applied, just like the two components of the calculation formula. This is evident from all comparisons between the cropping systems tested, taken two by two.

The comparative study between the conventional and the subsistence system presented a number of aspects concerning the productive potential (yield), the quality of the production (test weight, protein content, wet gluten content, deformation index, sedimentation index, and gluten index), other elements that determine the yield (number of wheat ears/sqm, thousand kernel weight, number of grains/sqm, and grains’ weight/sqm), and the size of the wheat plant.

The gluten index and the gluten deformation index parameters cannot be mutually supportive in relation to the analytical quality assessment of crops [67]. The quality assessment of the wheat must include both quality parameters because neither of them covers aspects that are fully related to the quality of the gluten. The gluten deformation index is appropriate for expressing the proteolytic activity, and the gluten index, especially at high values, expresses the native qualities of gluten [67,68].

In summary, the results are presented in Table 14.

## 3. Materials and Methods

### 3.1. Characteristics of the Experimental Site and Experimental Design

The experiments have been set for the period 2020–2022, at the University of Craiova—nonteaching department S.C.D.A. Caracal (Romania). The Caracal region is located in the south of the country (coordinates: 44°06′45″ N 24°20′50″ E), on the plains between the lower parts of the Jiu and Olt rivers. The region’s plains are well known for their agricultural specialty in cultivating grains.

The experiment was carried out on a typical argic (noncarbonic) chernoziomic soil, with a well-defined profile and insignificant differences regarding the physical, hydric, and chemical properties. The chernozems in general and the argic ones equally represent soils with a bioenergetic potential and a good production capacity.

The soil has a medium content of humus in the arable layer (2.20%), it is poorly supplied with nitrogen (0.104 N total), medium to well supplied with phosphorus (47 ppm mobile P), and well to very well supplied with potassium (K mobile 244.5 ppm), and pH_H2O_ has a value of 5.40.

The climatic characteristics of the studied region are presented in Table 15. In each of the test years, the average temperature for the growing season of wheat was above normal temperatures at Caracal. In terms of rainfall, the first two agricultural years showed a surplus, especially in 2020–2021. The precipitation that fell in May, in the full process of filling the grain, in each of the years helped considerably to obtain high yields. The yield achieved in the summer of 2020 was the highest yield achieved in the entire history of the location, which spans a period of 60 years. Although in the 2021–2022 agricultural year, the amount of water was reduced compared to the multi-year average, the precipitation that fell in April and May helped considerably in the formation of yield.

The experiments were located as follows (Figure 3):

Experiment 1—two-factor experiment in conventional system (differential fertilization technologies) where:

Factor A—cultivar with 25 graduations (Table 16).

Factor B—fertilization level with 2 graduations: N_40_P_40_ (40 Kg/ha of N and 40 Kg/ha of P_2_O_5_) (conv low-input) and N_100_P_40_ (100 Kg/ha of N and 40 Kg/ha of P_2_O_5_) (conv).

Experiment 2—two-factor experiment in conventional system (sown in late):

Factor A—cultivar with 25 graduations (Table 16).

Factor B—sowing season with 2 graduations: sown in the first half of October (conv) and sown at the end of October (conv delayed).

Experiment 3—single-plant experiment in subsistence system (subsistence):

The same 25 cultivars of autumn wheat sown under nonfertility conditions, without treatment against diseases and pests and harvested with a sickle.

Since the two experiments in conventional and subsistence systems had the same composition of tested cultivars, in the end, the data were also interpreted comparatively in the form of datasets.

The chosen cultivars were the Romanian and foreign cultivars of winter wheat, some of them with the largest expansion in the area but also nationally. The two-factor experiment was located in subdivided plots with 2 factors (25 variants × 3 replications). The single-factor experiment was located in a triple-balanced grid without repeating the basic scheme (25 variants × 3 replications). The harvestable area of each plot was 9 m^2^ (1.5 m wide and 6 m long).

### 3.2. Observations and Determinations

Field and laboratory determinations of yield and quality were carried out on each of the cultivars tested. The following were determined: number of wheat ears/sqm, plant height (cm), number of grains/wheat ear, weight of grains/ear (expressed in grams), grain yield (expressed in q/ha), thousand-kernel weight (TKW—grams), test weight (TW—kg/hl), protein content (Pr%), wet gluten content (WGC%), Zeleny sedimentation index (Zel expressed in mL), deformation index (D expressed in mm), and gluten index (GI determined by calculation = WGC (2 − D × 0.065).

### 3.3. Statistical Analysis

The Mann–Whitney U test was used to calculate the difference between two systems placed in different experiments [69]. Interpretation of the results was performed using the smallest value of U between U1 and U2, as follows:−If U ≤ U critical → significant results;−If U> U critical → nonsignificant results.

Verification key: U1 + U2 must equal n1 × n2. The sum of the hierarchies of the two datasets must equal the product of the number of the two datasets (in our case—625).

This test allows the comparison of only two data series, based on the ranking of the obtained values and then on their total added value. The lowest value was scored with 1 point and the highest value with 50 points, with 25 values in each series.

One-way ANOVA [69] was used to calculate the difference between multiple datasets. The calculations are based on the formulas shown in Table 17.

To limit the influence of the cultivars tested, the calculation was performed for the five highest values from each cropping system. The variation between systems reflects the variability between the data of the different series analyzed. Variation within cropping system reflects only the variability of cultivars. If the applied technology has no effect, the between-system and within-system variance should be similar, and the F-value should be close to 1.

There are two degrees of freedom associated with the F-value: one associated with the between-system variance (4 − 1 = 3) and one associated with the within-system variance (4 systems × 5 values each = 20 − 4 = 16).

From Toby Carter’s table for the critical values of the F-distribution, it was found that the value to be reported is 3.239 (interaction of degrees of freedom 3 and 16) [69].

Thus, the results obtained came from comparisons of:Between the 4 systems tested (low-input conventional, conventional, delayed conventional, and subsistence) with the help of the one-way ANOVA test, a test that allows comparison of multiple sets of data.Between low-input conventional and conventional and between conventional and delayed conventional using the limit difference (LD) calculation program PS2F (2 factor subdivided plots) where for the first comparison, the factor A—cultivar had 25 gradations and factor B—fertilization level had 2 gradations: N_40_P_40_ and N_100_P_40_, in 3 replications, with cultivar mean as control; and for the second comparison—cultivar had 25 graduations and factor B—sowing time had 2 graduations: sown on 15 October and sown on 30 October, in 3 replications, with the cultivar mean as control;Between low-input conventional and subsistence and between conventional and subsistence, based on the Mann–Whitney U test. For the first comparison, we considered the systems to be close in terms of fertilization level (low nitrogen dose vs. unfertilized). For the second comparison, we considered the systems to be at opposite poles of each other (high nitrogen dose vs. unfertilized).

The comparison of the systems two by two was undertaken to highlight the nature of the differences (significant or nonsignificant) and which technologies determine them.

Similar studies in terms of crop management level (low input and high input) were conducted in Poland in 3 locations in 2009 and 2010 on 7 wheat cultivars [56].

## 4. Conclusions

We can state that cropping systems influence all the studied elements except the thousand-kernel weight. The superiority of the delayed conventional system was shown by the number of grains/wheat ear and the deformation index.

Protein content was differentiated between the conventional and subsistence system variants but especially between low-input conventional and conventional. The obtaining of a high-quality wheat yield relies on nitrogen supply. The fact that three quality indicators (protein content, wet gluten content, and gluten index) are very significantly higher when the nitrogen dose is increased strongly supports the above statement.

Given that 25 wheat cultivars of different origins were tested over three years, the results are conclusive enough and give scientific accuracy to this study.

What is important to note is that the productivity and proteins are positively correlated under certain conditions, but this requires further in-depth studies.

## Figures and Tables

**Figure 1 plants-12-02802-f001:**
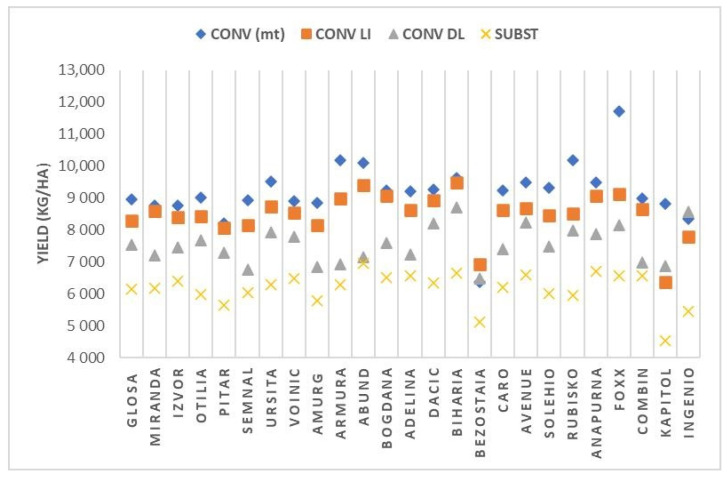
Yield of tested cultivars according to the applied technological management (average 2020–2022).

**Figure 2 plants-12-02802-f002:**
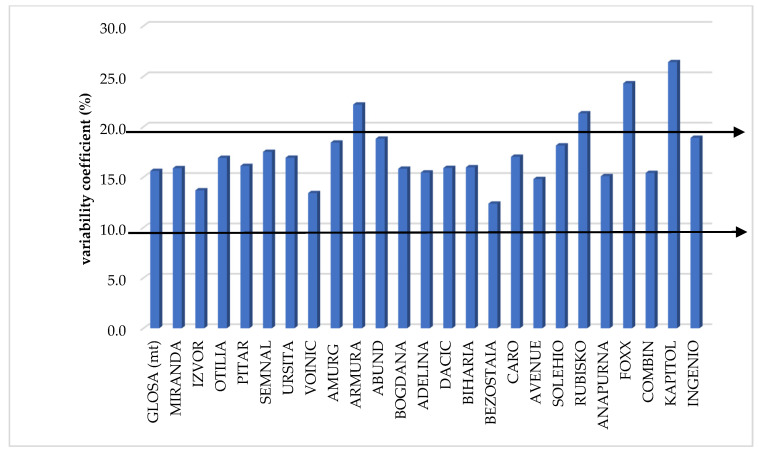
Coefficient of variability (CV%) of production according to the technological management applied.

**Figure 3 plants-12-02802-f003:**
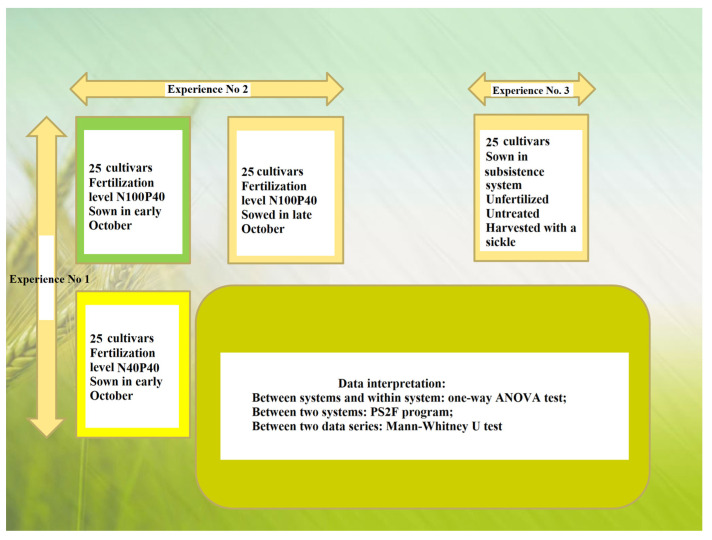
The design of experiments with wheat at SCDA Caracal.

**Table 1 plants-12-02802-t001:** Technological management influence on number of wheat ears/sqm—Caracal, 2020–2022 average.

**I. One-way ANOVA test, multiple comparisons**						
**Source of variance**	**Sum of squares**		**Degrees of freedom**		**Variance**	**F**
Between systems	247,695		3		82,565	126.686
Within systems	10,427.68		16		651.73	
Total	258,122.7		19			
**II. Two systems compared using SP2F (subdivided plots with 2 factors) program**						
**Technological management**	**Number of wheat ears/sqm**		**Difference**		**Significance**	
conv low-input	694		Ct			
conv	743		+49			
LD 5% = 132 ears/sqm; LD 1% = 180 ears/sqm; LD 0.1% = 245 ears/sqm						
conv	743		Ct			
conv delayed	502		−241		ooo	
LD 5% = 53 ears/sqm; LD 1% = 70 ears/sqm; LD 0.1% = 92 ears/sqm						
**III. Difference between two samples of data using Mann–Whitney U test**						
**Technological management**	**Sample size**	**Number of wheat ears/sqm**	**Mean rank**	**Sum of ranks (R_1_ + R_2_)**	**Significance**	
conv low-input	25	694	37.96	949	U_(1)_ < U_critical(211)_	
subsistence	25	471	13.04	326	*p* < 0.05	
Total (n_1_ + n_2_)	50					
conv	25	743	37.62	940.5	U_(9.5)_ < U_critical(211)_	
subsistence	25	471	13.38	334.5	*p* < 0.05	
Total (n_1_ + n_2_)	50					
LD	Significance			Positive	Negative	
5–1%	Significant			*	o	
1–0.1%	Distinctly significant			**	oo	
>0.1%	Very significant			***	ooo	

**Table 2 plants-12-02802-t002:** Technological management influence on plant height—Caracal, 2020–2022 average.

**I. One-way ANOVA test, multiple comparisons**						
**Source of variance**	**Sum of squares**		**Degrees of freedom**		**Variance**	**F**
Between systems	1210.84		3		403.613	20.157
Within systems	320.35		16		20.022	
Total	1531.17		19			
**II. Two systems compared using SP2F (subdivided plots with 2 factors) program**						
**Technological management**	**Plant height (cm)**		**Difference**		**Significance**	
conv low-input	89.4		Ct			
conv	89.8		+0.4			
LD 5% = 6.4 cm; LD 1% = 8.8 cm; LD 0.1% = 11.9 cm						
conv	89.8		Ct			
conv delayed	86.4		−3.4			
LD 5% = 4 cm; LD 1% = 6 cm; LD 0.1% = 9 cm						
**III. Difference between two samples of data using Mann–Whitney U test**						
**Technological management**	**Sample size**	**Plant height (cm)**	**Mean rank**	**Sum of ranks (R_1_ + R_2_)**	**Significance**	
conv low-input	25	89.4	33.02	825.5	U_(124.5)_ < U_critical(211_	
subsistence	25	81.6	17.98	449.5	*p* < 0.05	
Total	50					
conv	25	89.8	32.1	802.5	U_(147.5)_ < U_critical(211)_	
subsistence	25	81.6	18.9	472.5	*p* < 0.05	
Total	50					

**Table 3 plants-12-02802-t003:** Technological management influence on number of grains/wheat ear– Caracal, 2020–2022 average.

**I. One-way ANOVA test, multiple comparisons**						
**Source of variance**	**Sum of squares**		**Degrees of freedom**		**Variance**	**F**
Between systems	135		3		45	7.129
Within systems	101		16		6.31	
Total	236		19			
**II. Two systems compared using SP2F (subdivided plots with 2 factors) program**						
**Technological management**	**Number of grains/wheat ear**		**Difference**		**Significance**	
conv low-input	40		Ct			
conv	40		0			
LD 5% = 5 grains no/ear; LD 1% = 7 grains no/ear; LD 0.1% = 10 grains no/ear						
conv	40		Ct			
conv delayed	44		+4		*	
LD 5% = 4 grains no/ear; LD 1% = 7 grains no/ear; LD 0.1% = 10 grains no/ear						
**III. Difference between two samples of data using Mann–Whitney U test**						
**Technological management**	**Sample size**	**Number of grains/wheat ear**	**Mean rank**	**Sum of ranks (R_1_ + R_2_)**	**U_1_ and U2**	
conv low-input	25	40	29.12	728	U_(222)_ > U_critical(211)_	
subsistence	25	38	21.88	547	*p* > 0.05	
Total	50					
conv	25	40	28.86	721.5	U_(228.5)_ > U_critical(211)_	
subsistence	25	38	22.14	553.5	*p* > 0.05	
Total	50					

**Table 4 plants-12-02802-t004:** Technological management influence on weight of grains/ear–Caracal, 2020–2022 average.

**I. One-way ANOVA test, multiple comparisons**						
**Source of variance**	**Sum of squares**		**Degrees of freedom**		**Variance**	**F**
Between systems	0.034		3		0.011333	4.90
Within systems	0.037		16		0.002313	
Total	0.071		19			
**II. Two systems compared using SP2F (subdivided plots with 2 factors) program**						
**Technological management**	**Weight of grains/ear (g)**		**Difference**		**Significance**	
conv low-input	1.64		Ct			
conv	1.60		−0.04			
LD 5% = 0.32 g; LD 1% = 0.49 g; LD 0.1% = 0.50 g						
conv	1.60		Ct			
conv delayed	1.56		−0.04			
LD 5% = 0.28 g; LD 1% = 0.41 g; LD 0.1% = 0.52 g						
**III. Difference between two samples of data using Mann–Whitney U test**						
**Technological management**	**Sample size**	**Weight of grains/ear (g)**	**Mean rank**	**Sum of ranks (R_1_ + R_2_)**	**Significance**	
conv low-input	25	1.64	31.74	793.5	U_(156.5)_ < U_critic(211)_	
subsistence	25	1.54	19.26	481.5	*p* < 0.05	
Total	50					
conv	25	1.60	29.06	726.5	U_(223.5)_ > U_critic(211)_	
subsistence	25	1.54	21.94	548.5	*p* > 0.05	
Total	50					

**Table 5 plants-12-02802-t005:** Technological management influence on yield—Caracal, 2020–2022 average.

**I. One-way ANOVA test, multiple comparisons**						
**Source of variance**	**Sum of squares**		**Degrees of freedom**		**Variance**	**F**
Between systems	3451.16		3		1150.39	58.86
Within systems	312.69		16		19.54	
Total	3763.85		19			
**II. Two systems compared using SP2F (subdivided plots with 2 factors) program**						
**Technological management**	**Yield (q/ha)**		**Difference**		**Significance**	
conv low-input	84.85		Ct			
conv	91.77		+6.92			
LD 5% = 7.0 q/ha; LD 1% = 9.6 q/ha; LD 0.1% = 13 q/ha						
conv	91.77		Ct			
conv delayed	75.37		−16.40		oo	
LD 5% = 10.3 q/ha; LD 1% = 15.8 q/ha; LD 0.1% = 19.8 q/ha						
**III. Difference between two samples of data using Mann–Whitney U test**						
**Technological management**	**Sample size**	**Yield (q/ha)**	**Mean rank**	**Sum of ranks (R_1_ + R_2_)**	**Significance**	
conv low-input	25	84.85	37.56	939	U_(11)_ < U_critical(211)_	
subsistence	25	61.61	13.44	336	*p* < 0.05	
Total	50					
conv	25	91.77	37.6	940	U_(10)_ < U_critical(211)_	
subsistence	25	61.61	13.4	335	*p* < 0.05	
Total	50					

**Table 6 plants-12-02802-t006:** Coefficient of variability of the production obtained during each year of the testing.

No. Crt.	Cultivar	CV between the Years of Experimentation (%)
1	GLOSA	13.43
2	MIRANDA	17.30
3	IZVOR	3.40
4	OTILIA	13.96
5	PITAR	22.77
6	SEMNAL	16.08
7	URSITA	13.30
8	VOINIC	10.75
9	AMURG	18.35
10	ARMURA	7.18
11	ABUND	20.59
12	BOGDANA	24.19
13	ADELINA	10.25
14	DACIC	17.87
15	BIHARIA	21.74
16	BEZOSTAIA	6.69
17	CARO	18.67
18	AVENUE	26.81
19	SOLEHIO	26.73
20	RUBISKO	28.22
21	ANAPURNA	25.37
22	FOXX	26.05
23	COMBIN	4.80
24	KAPITOL	26.04
25	INGENIO	27.22

**Table 7 plants-12-02802-t007:** Technological management influence on thousand-kernel weight (TKW)—Caracal, 2020–2022 average.

**I. One-way ANOVA test, multiple comparisons**						
**Source of variance**	**Sum of squares**		**Degrees of freedom**		**Variance**	**F**
Between systems	8.08		3		2.693333	1.363
Within systems	31.62		16		1.97625	
Total	39.70		19			
**II. Two systems compared using SP2F (subdivided plots with 2 factors) program**						
**Technological management**	**TKW (g)**		**Difference**		**Significance**	
conv low-input	39.4		Ct			
conv	37.8		−1.6			
LD 5% = 4.5 g; LD 1% = 6.2 g; LD 0.1% = 8.3 g						
conv	37.8		Ct			
conv delayed	36.2		−1.6			
LD 5% = 2.8 g; LD 1% = 4.2 g; LD 0.1% = 6.4 g						
**III. Difference between two samples of data using Mann–Whitney U test**						
**Technological management**	**Sample size**	**TKW (g)**	**Mean rank**	**Sum of ranks (R_1_ + R_2_)**	**U_1_ and U2**	
conv low-input	25	39.4	29.44	736	U_(214)_ > U_critical(211)_	
subsistence	25	38.0	21.56	539	*p* > 0.05	
Total	50					
conv	25	37.8	25.6	640	U_(300)_ > U_critical(211)_	
subsistence	25	38.0	25.0	625	*p* > 0.05	
Total	50					

**Table 8 plants-12-02802-t008:** Technological management influence on test weight (TW)—Caracal, 2020–2022 average.

**I. One-way ANOVA test, multiple comparisons**						
**Source of variance**	**Sum of squares**	**Degrees of freedom**		**Variance**		**F**
Between systems	171.93	3		57.3		29.11
Within systems	31.46	16		2.0		
Total	203.39	19				
**II. Two systems compared using SP2F (subdivided plots with 2 factors) program**						
**Technological management**	**TW (kg/hl)**	**Difference**		**Significance**		
conv low-input	75.8	Ct				
conv	72.2	−3.6		oo		
LD 5% = 2.3 kg/hl; LD 1% = 3.1 kg/hl; LD 0.1% = 4.1 kg/hl						
conv	72.2	Ct				
conv delayed	66.7	−5.5		ooo		
LD 5% = 1.8 kg/hl; LD 1% = 2.6 kg/hl; LD 0.1% = 4.2 kg/hl						
**III. Difference between two samples of data using Mann–Whitney U test**						
**Technological management**	**Sample size**	**TW (kg/hl)**	**Mean rank**	**Sum of ranks (R_1_ + R_2_)**	**Significance**	
conv low-input	25	75.8	31.68	792	U_(158)_ < U_critical(211)_	
subsistence	25	73.1	19.32	483	*p* < 0.05	
Total	50					
conv	25	72.2	31.18	779.5	U_(170.5)_ < U_critical(211)_	
subsistence	25	73.1	19.82	495.5	*p* < 0.05	
Total	50					

**Table 9 plants-12-02802-t009:** Technological management influence on protein content—Caracal, 2020–2022 average.

**I. One-way ANOVA test, multiple comparisons**						
**Source of variance**	**Sum of squares**		**Degrees of freedom**		**Variance**	**F**
Between systems	11.76		3		3.92	24.123
Within systems	2.6		16		0.1625	
Total	14		19			
**II. Two systems compared using SP2F (subdivided plots with 2 factors) program**						
**Technological management**	**Protein content (%)**		**Difference**		**Significance**	
conv low-input	10.8		Ct			
conv	13.0		+2.2		***	
LD 5% = 0.5%; LD 1% = 0.8%; LD 0.1% = 1%						
conv	13.0		Ct			
conv delayed	13.0		0			
LD 5% = 1.3%; LD 1% = 2.4%; LD 3.8% = 1%						
**III. Difference between two samples of data using Mann–Whitney U test**						
**Technological management**	**Sample size**	**Protein content (%)**	**Mean rank**	**Sum of ranks (R_1_ + R_2_)**	**U_1_ and U2**	
conv low-input	25	10.8	25.28	632	U_(307)_ > U_critical(211)_	
subsistence	25	10.0	25.72	643	*p* > 0.05	
Total	50					
conv	25	13.0	30.9	772.5	U_(177.5)_ < U_critical(211)_	
subsistence	25	10.0	20.1	502.5	*p* < 0.05	
Total	50					

**Table 10 plants-12-02802-t010:** Technological management influence on wet gluten content—Caracal, 2020–2022 average.

**I. One-way ANOVA test, multiple comparisons**						
**Source of variance**	**Sum of squares**		**Degrees of freedom**		**Variance**	**F**
Between systems	44.3		3		14.77	46.693
Within systems	5.06		16		0.32	
Total	50.34		19			
**II. Two systems compared using SP2F (subdivided plots with 2 factors) program**						
**Technological management**	**Wet gluten content (%)**		**Difference**		**Significance**	
conv low-input	21.5		Ct			
conv	27.9		+6.4		***	
LD 5% = 1.8%; LD 1% = 2.4%; LD 0.1% = 3.1%						
conv	27.9		Ct			
conv delayed	23.2		−4.7		oo	
LD 5% = 3.4%; LD 1% = 4.5%; LD 0.1% = 5.9%						
**III. Difference between two samples of data using Mann–Whitney U test**						
**Technological management**	**Sample size**	**Wet gluten content (%)**	**Mean rank**	**Sum of ranks (R_1_ + R_2_)**	**Significance**	
conv low-input	25	21.5	28.56	714	U_(236)_ > U_critical(211)_	
subsistence	25	22.0	22.44	561	*p* > 0.05	
Total	50					
conv	25	27.9	30.76	769	U_(181)_ < U_critical(211)_	
subsistence	25	22.0	20.24	506	*p* < 0.05	
Total	50					

**Table 11 plants-12-02802-t011:** Technological management influence on the Zeleny index—Caracal, 2020–2022 average.

**I. One-way ANOVA test, multiple comparisons**						
**Source of variance**	**Sum of squares**		**Degrees of freedom**		**Variance**	**F**
Between systems	495		3		165	67.9
Within systems	38.87		16		2.43	
Total	509.7		19			
**II. Two systems compared using SP2F (subdivided plots with 2 factors) program**						
**Technological management**	**Zeleny index (mL)**		**Difference**		**Significance**	
conv low-input	73		Ct			
conv	71		−2			
LD 5% = 2.4 mL; LD 1% = 3.2 mL; LD 0.1% = 4.2 mL						
conv	71		Ct			
conv delayed	64		−7		ooo	
LD 5% = 4.2 mL; LD 1% = 5.4 mL; LD 0.1% = 7.0 mL						
**III. Difference between two samples of data using Mann–Whitney U test**						
**Technological management**	**Sample size**	**Zeleny index (mL)**	**Mean rank**	**Sum of ranks (R_1_ + R_2_)**	**Significance**	
conv low-input	25	73	27.04	669.5	U_(280.5)_ > U_critical(211)_	
subsistence	25	72	24.22	605.5	*p* > 0.05	
Total	50					
conv	25	71	27.04	676	U_(274)_ > U_critical(211)_	
subsistence	25	72	24.36	609	*p* > 0.05	
Total	50					

**Table 12 plants-12-02802-t012:** Technological management influence on deformation index—Caracal, 2020–2022 average.

**I. One-way ANOVA test, multiple comparisons**						
**Source of variance**	**Sum of squares**		**Degrees of freedom**		**Variance**	**F**
Between systems	595		3		198.33	136.544
Within systems	23.24		16		1.4525	
Total	625.24		19			
**II. Two systems compared using SP2F (subdivided plots with 2 factors) program**						
**Technological management**	**Deformation index (mL)**		**Difference**		**Significance**	
conv low-input	7		Ct			
conv	4		−3		ooo	
LD 5% = 1.3 mm; LD 1% = 1.8 mm; LD 0.1% = 2.3 mm						
conv	4		Ct			
conv delayed	8		+4		**	
LD 5% = 3.0 mm.; LD 1% = 4.0 mm; LD 0.1% = 5.0 mm						
**III. Difference between two samples of data using Mann–Whitney U test**						
**Technological management**	**Sample size**	**Deformation index (mL)**	**Mean rank**	**Sum of ranks (R_1_ + R_2_)**	**Significance**	
conv low-input	25	7	18.34	458.5	U_(133.5)_ < U_critical(211)_	
subsistence	25	12	32.66	816.5	*p* < 0.05	
Total	50					
conv	25	4	16.28	407	U_(82)_ < U_critical(211)_	
subsistence	25	12	34.72	868	*p* < 0.05	
Total	50					

**Table 13 plants-12-02802-t013:** Technological management influence on gluten index—Caracal, 2020–2022 average.

**I. One-way ANOVA test, multiple comparisons**						
**Source of variance**	**Sum of squares**		**Degrees of freedom**		**Variance**	**F**
Between systems	164.85		3		54.95	12.690
Within systems	69.28		16		4.33	
Total	234.13		19			
**II. Two systems compared using SP2F (subdivided plots with 2 factors) program**						
**Technological management**	**Gluten index (%)**		**Difference**		**Significance**	
conv low-input	33.2		Ct			
conv	48.5		+15.3		***	
LD 5% = 3.5%; LD 1% = 4.7%; LD 0.1% = 6.1%						
conv	48.5		Ct			
conv delayed	34.3		−14.2		ooo	
LD 5% = 7%; LD 1% = 9%; LD 0.1% = 11%						
**III. Difference between two samples of data using Mann–Whitney U test**						
**Technological management**	**Sample size**	**Gluten index (%)**	**Mean rank**	**Sum of ranks (R_1_ + R_2_)**	**Significance**	
conv low-input	25	33.2	30.86	771.5	U_(178.5)_ < U_critical(211)_	
subsistence	25	26.8	20.14	503.5	*p* < 0.05	
Total	50					
conv	25	48.5	33.78	844.5	U_(105.5)_ < U_critical(211)_	
subsistence	25	26.8	17.22	430.5	*p* < 0.05	
Total	50					

**Table 14 plants-12-02802-t014:** Synthesis of comparative results on wheat crops through the view of the differentiated technological management applied *.

Analyzed Trait	Comparative Results				
	Between All Systems	Conv Low-Input–Conv	Conv–Conv Delayed	Conv Low-Input–Subsistence	Conv–Subsistence
Number of wheat ears/sqm	Significant differences	Nonsignificant differences	ooo _(Very significant negative)_	Significant differences	Significant differences
Plant height	Significant differences	Nonsignificant differences	Nonsignificant differences	Significant differences	Significant differences
Number of grains/ear	Significant differences	Nonsignificant differences	* _(Significant positive)_	Nonsignificant differences	Nonsignificant differences
Weight of grains/wheat ear	Significant differences	Nonsignificant differences	Nonsignificant differences	Significant differences	Significant differences
Grain yield	Significant differences	Nonsignificant differences	oo _(Distinctly significant negative)_	Significant differences	Significant differences
Test weight	Significant differences	oo_(Distinctly significant negative)_	ooo _(Very significant negative)_	Significant differences	Significant differences
Thousand kernel weight	Nonsignificant differences	Nonsignificant differences	Nonsignificant differences	Nonsignificant differences	Nonsignificant differences
Protein content	Significant differences	*** _(Very significant positive)_	Nonsignificant differences	Nonsignificant differences	Significant differences
Wet gluten content	Significant differences	*** _(Very significant positive)_	oo _(Distinctly significant negative)_	Nonsignificant differences	Significant differences
Zeleny index	Significant differences	Nonsignificant differences	ooo _(Very significant negative)_	Nonsignificant differences	Nonsignificant differences
Deformation index	Significant differences	ooo _(Very significant negative)_	** _(Distinctly significant positive)_	Significant differences	Significant differences
Gluten index	Significant differences	*** _(Very significant positive)_	ooo _(Very significant negative)_	Significant differences	Significant differences

* The meaning of colors: yellow—characters that showed significant differences obtained by one-way ANOVA and Mann–Whitney methods; red—the characters that showed decreases with statistical assurance obtained through the SP2F program; blue—characters that showed increases with statistical assurance obtained through the SP2F program; no color—characters that were not influenced by the applied management (interpretation performed after calculation by all three methods).

**Table 15 plants-12-02802-t015:** The climatic characteristics of the studied region (average 2020–2022).

Specification	October	November	December	January	February	March	April	May	June	Total/ Agricultural Year	Average/ Agricultural Year
Temperature °C											
2020	13.6	9.9	3.2	0.8	5.6	7.5	11.9	16.8	20.8		10.0
2021	14	5.0	3.2	1.5	3.2	5.0	9.7	17.4	21.4		8.9
2022	10.2	7.4	2.6	2.0	4.2	4.3	11.1	18.1	23.0		9.2
Normal	11.7	5.1	0.3	−1.3	0.8	6.0	12.0	17.7	21.6		8.2
Rainfall (mm)											
2020	20.8	102.4	25.4	8.4	47.4	49.4	12.8	61.6	108	436.2	
2021	46.2	19.6	70.4	98.0	29.6	92.4	32.6	55.6	103.2	547.6	
2022	101.4	28.0	60.8	19.2	4.8	13.2	77.8	44.6	14.2	364.0	
Multiannual average	46.0	37.0	39.1	30.8	26.3	34.2	47.8	58.6	69.7	389.5	
	Relative humidity %										
2020	73.9	91.0	92.0	83.5	76.5	75.4	52.9	67.3	77.0		76.6
2021	83.6	95.0	98.8	95.4	85.6	80.8	73.2	69.5	79.7		84.6
2022	82.8	93.7	94.7	84.3	74.5	68.2	77.2	68.1	69.1		79.2

**Table 16 plants-12-02802-t016:** Tested wheat cultivars.

Factor A	The Cultivar	Provenance
1	Glosa	Romania–Fundulea
2	Miranda	Romania–Fundulea
3	Otilia	Romania–Fundulea
4	Pitar	Romania–Fundulea
5	Semnal	Romania–Fundulea
6	Ursita	Romania–Fundulea
7	Voinic	Romania–Fundulea
8	Abund	Romania–Fundulea
9	Izvor	Romania–Fundulea
10	Caro	Romania–Caracal
11	Adelina	Romania–Șimnic
12	Dacic	Romania–Lovrin
13	Biharia	Romania–Lovrin
14	Amurg	Romania–Fundulea
15	Armura	Romania–Fundulea
16	Bogdana	Romania–Fundulea
17	Bezostaia	Russia
18	Avenue	LG
19	Anapurna	LG
20	Rubisko	RAGT
21	Solehio	KWS
22	Foxx	BIOCROP
23	Combin	BIOCROP
24	Moisson	SYNGENTA
25	Ingenio	SYNGENTA

**Table 17 plants-12-02802-t017:** Calculation of F statistic for an ANOVA *.

Source (of Variance)	Sum of Squares	Degrees of Freedom	Mean Square (or Variance)	F
Between data sets	∑n($\bar{y}$ − $\bar{G}$)^2^	k − 1	column 2/column 3	row 2/row 3
Within data set (error)	∑(y − $\bar{y}$)^2^	N − k	column 2/column 3	
Total	∑(y − $\bar{G}$)^2^	N − 1		

* N = total sample size (the sum of the sample sizes for each data set); k = number of agricultural systems; $\bar{G}$ = the grand mean (the mean of the combined data from all datasets); $\bar{y}$ = a dataset mean; n = a sample size; ∑n($\bar{y}$ − $\bar{G}$)^2^ = the sum of the squared differences between each dataset mean and the grand mean multiplied by the sample size of that dataset; ∑(y − $\bar{y}$)^2^ = the sum of the squared differences between each y and its dataset mean; ∑(y − $\bar{G}$)^2^ = the sum of the squared differences between each y and the grand mean; if F ≥ F critical → significant result; if F < F critical → nonsignificant result.

## Data Availability

The data presented in this study are available on request from the corresponding author.

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
