# Peer review of "The Effect of Different Cropping Systems on Yield, Quality, Productivity Elements, and Morphological Characters in Wheat (*Triticum aestivum*)"

_plants, 2023, doi:10.3390/plants12152802_

Round 1

Reviewer 1 Report

The paper entitled ‘Results Regarding the Influence of Some Different Cropping Systems on the Yield, Quality, Productivity Elements and Morphological Characters in Wheat (Triticum aestivum) (Plants-2496703) regards an overview on the effect of different cropping systems on morphology, yield and quality. A lack of discussion was evidenced, also in the light of climatic conditions. A revision of English is required. Also, Tables needed a revision for various mistakes and improve making them more presentable.

Although the topic is of interest, it cannot be accepted in this form but needed major revisions.

Title

The title should be revised as follows “The effect of different cropping systems on yield, quality, productivity elements and morphological characters in wheat (Triticum aestivum)”.

Abstract

According to the instructions for authors ‘The abstract should be a single paragraph and should follow the style of structured abstracts, but without headings: 1) Background: Place the question addressed in a broad context and highlight the purpose of the study; 2) Methods: Describe briefly the main methods or treatments applied. Include any relevant preregistration numbers, and species and strains of any animals used; 3) Results: Summarize the article's main findings; and 4) Conclusion: Indicate the main conclusions or interpretations. The abstract should be an objective representation of the article: it must not contain results which are not presented and substantiated in the main text and should not exaggerate the main conclusions’. The abstract here reported is unbalanced as it contains too many details of the experimental site and treatments that should be in the Materials and methods section and few results. Please, check carefully.

Keywords

Check the keywords, in particular ‘traits’. It should be better to introduce morphology and change management in technological management.

Introduction

Line 57. What is the logical sequence between the ‘security’ of line 57 with the ‘calories’ of the following lines? Moreover, wheat is one of the three main cereals. Which are the others?

Line 79. It is not clear the difference between the ‘farmer practice’ respect to the ‘high-input’ approach.

Line 104. Clarify better the expression ‘subsistence farming’ in the aim of the work.  Moreover, move on ‘productivity elements’ after yield.

Results and Discussion

Section 2.1. Although this section is within the Results and Discussion one, the authors reported mainly the results without proper discussion and references to the literature in the field. Please, improve this section.

Section 2.2. Lines 123-126. The discussion should be reported after the results in order to be interpreted in perspective of previous studies. Moreover, the discussion reported in the lines 123-126 refers to the coefficient of variation as a good criterion for height stability, but this trait has not been considered by the authors in the manuscript. Review carefully.

Section 2.3. Clarify better the expression ‘semi-significant’ difference. Also, uniform the significance among the Tables (sometimes expressed with number and others with asterisk). The discussion reported in lines 154-155 requires to be implemented.

Section 2.4. No discussion has been reported.

Section 2.5. The text reported in lines 174-178 is to be revisited. Also check lines 184-207, sometimes confusing.

Lines 222-229. Check the English language. As the authors in these lines and previously (see section 2.2) mentioned the coefficient of variation (CV), the authors should show data of CV in a Table. Moreover, the authors should link better the two periods reported in lines 222-225 and 226-229 in order to define the best stable cultivars in terms of yield evidenced in lines 230-234.

Lines 231- 232. The authors justify the different performance of foreign varieties to be strongly affected by “Climatic conditions”. No comment was reported by the authors with a view to interpreting the yield and yield-related traits in the light of climatic conditions reported in table 14. The same consideration is applied to all traits.

Section 2.7. The discussion should be reported after the results in order to be interpreted in perspective of previous studies. Clarify better this section and check the English.

Section 2.8. Line 272. Please, report firstly the results and then discussions. As authors stated that ‘earlier studies showed that an increase in yield decreases protein content’, some references should be added. Line 273. The result reported in reference 47 was limited to the two winter wheat varieties analyzed in that study, so the authors should add a clarification like ‘for genotype-specific’. Write better the period reported in lines 281-286, particularly the last sentence.

Section 2.9. Check the expression ‘culture system’. No discussion was reported for this trait.

Section 2.10. Check English and add discussion.

Section 2.11. No discussion. English required.

Section 2.12. No discussion. English required.

As in Table 13 is reported a summary of different management technologies applied without a comment, the same could have been moved to the supplementary materials.

Tables should be reviewed, avoiding orthographic mistakes and uniforming the significance (sometimes reported with asterisk and sometimes with zero number).

Materials and methods

In Table 14, no multiannual average was reported for relative humidity. Please, substitute Roman numbers with months.

Conclusions

The conclusions are confusing and should be reviewed, in particular highlighting that productivity and proteins are positively correlated under certain conditions and that this requires further in-depth studies

Check the expression: very significant differences line 466; statistically inferior line 473, very significantly higher lines 482-483.

Improve the readability of your paper

Author Response

Response to Reviewer 1 Comments

Dear Reviewer,

First of all, we thank you for accepting the revision of our manuscript as well as for all the comments and suggestions provided.

Point 1. The paper entitled ‘Results Regarding the Influence of Some Different Cropping Systems on the Yield, Quality, Productivity Elements and Morphological Characters in Wheat (Triticum aestivum)’ (Plants-2496703)’ regards an overview on the effect of different cropping systems on morphology, yield and quality. A lack of discussion was evidenced, also in the light of climatic conditions. A revision of English is required. Also, Tables needed a revision for various mistakes and improve making them more presentable.

Although the topic is of interest, it cannot be accepted in this form but needed major revisions.

Response 1: In the revised version of the manuscript we have considered each of these suggestions. We have proofread the English throughout the manuscript.

Point 2. Title

The title should be revised as follows “The effect of different cropping systems on yield, quality, productivity elements and morphological characters in wheat (Triticum aestivum)”

Response 2: Thank you for the suggestion. The title has been changed accordingly.

Point 3. Abstract

According to the instructions for authors ‘The abstract should be a single paragraph and should follow the style of structured abstracts, but without headings: 1) Background: Place the question addressed in a broad context and highlight the purpose of the study; 2) Methods: Describe briefly the main methods or treatments applied. Include any relevant preregistration numbers, and species and strains of any animals used; 3) Results: Summarize the article's main findings; and 4) Conclusion: Indicate the main conclusions or interpretations. The abstract should be an objective representation of the article: it must not contain results which are not presented and substantiated in the main text and should not exaggerate the main conclusions’. The abstract here reported is unbalanced as it contains too many details of the experimental site and treatments that should be in the Materials and methods section and few results. Please, check carefully.

Response 3: Thank you for your suggestions. We have revised the Abstract section accordingly so that it now more clearly summarizes the central idea of the whole manuscript. We have removed details related to the experimental site and added some conclusive results.

Point 4. Keywords

Check the keywords, in particular ‘traits’. It should be better to introduce morphology and change management in technological management.

Response 4: We have modified the keywords as per your suggestion.

Point 5. Introduction

Point 5.1. Line 57. What is the logical sequence between the ‘security’ of line 57 with the ‘calories’ of the following lines? Moreover, wheat is one of the three main cereals. Which are the others?

Response 5.1: There are two distinct sentences here, in which we wanted, on the one hand, to stress the importance of wheat in ensuring food security, given that wheat is one of the most widely grown cereals, alongside rice and corn (we have now added this in the revised manuscript). On the other hand, we wanted to emphasise that since most wheat production is consumed in human nutrition, consumers supplement 60% of their daily calorie needs with wheat-based foods.

Point 5.2. Line 79. It is not clear the difference between the ‘farmer practice’ respect to the ‘high-input’ approach.

Response 5.2: We have taken into account your observation and defined the two terms in the revised form of the manuscript.

Point 5.3. Line 104. Clarify better the expression ‘subsistence farming’ in the aim of the work.  Moreover, move on ‘productivity elements’ after yield.

Response 5.3: Following your suggestion, we have clarified the term 'subsistence farming' in the paragraph regarding the purpose of our paper, giving some details to better express the essence of this type of agriculture. We have also moved 'productivity elements' after 'yield', as you have suggested, thank you.

Point 6. Results and Discussion

Section 2.1. Although this section is within the Results and Discussion one, the authors reported mainly the results without proper discussion and references to the literature in the field. Please, improve this section.

Response: Thank you for your comment. In the revised manuscript we have improved this section.

Section 2.2. Lines 123-126. The discussion should be reported after the results in order to be interpreted in perspective of previous studies. Moreover, the discussion reported in the lines 123-126 refers to the coefficient of variation as a good criterion for height stability, but this trait has not been considered by the authors in the manuscript. Review carefully.

Response: We have reported discussions immediately after the results, as you suggested, thank you.

Regarding the coefficient of variability, it is a simple bibliographical reference that wants to emphasize the role of plant size rather than yield as criteria for assessing stability. We wanted to show why we took into account the presentation of this characteristic and did not intend to present the coefficient of variability of the size. But in order not to create any confusion, we have improved the paper in this respect with a new table (Table 6) and a new figure (Figure 2).

Section 2.3. Clarify better the expression ‘semi-significant’ difference. Also, uniform the significance among the Tables (sometimes expressed with number and others with asterisk). The discussion reported in lines 154-155 requires to be implemented.

Response: That is correct, thank you for your observation. There is no such term as semi-significant it is only significant. It was an error that we fixed.

For tables, the numbers represent a difference which is a quantitative expression and the asterisk or circle represents a significance based on the fit within the LD values. Thus,

- the asterisk represents an increase: * -significant

                                                         ** - distinctly significant

                                                         ***-very significant

- the circle represents a decrease: o -significant

                                                     oo - distinctly significant

                                                     ooo-very significant

The discussion reported at lines 154-155 has been implemented as you suggested.

Section 2.4. No discussion has been reported.

Response: Thank you for your remark. In the revised manuscript we have added some discussion within this section.

Section 2.5. The text reported in lines 174-178 is to be revisited. Also check lines 184-207, sometimes confusing.

Response: We considered that if we tested 25 varieties we should also show some data on their yield and stability. Thus, we shifted from presenting the influence of the applied technology on yield to highlighting the varieties according to the results obtained through the coefficient of variability.

Lines 222-229. Check the English language. As the authors in these lines and previously (see section 2.2) mentioned the coefficient of variation (CV), the authors should show data of CV in a Table. Moreover, the authors should link better the two periods reported in lines 222-225 and 226-229 in order to define the best stable cultivars in terms of yield evidenced in lines 230-234.

Response: We have proofread the English throughout the manuscript.

We did not present in detail in order not to overload the paper, showing only what was essential. The first CV, as specified within the text, is calculated according to the management applied (between the 4 variants - low-input conventional, conventional, delayed conventional and subsistence). We have introduced a new figure in this regard for a better understanding of the text (Figure 2).

The second CV is calculated according to the yields obtained in each of the years of the experiment, regardless of the technology applied. We have introduced a new table for better understanding of the text (Table 6).

For linking better discussions on CV, as you recommended, we have removed this paragraph:

The coefficients of determination (r2) highlighted the fact that 67% of the variability in conventional yield is matched by variability in conventional low-input yield. Also, 57% of the variability in conventional yield is matched by variability in subsistence system yield but only 41% by variability in conventional delayed yield”.

Lines 231- 232. The authors justify the different performance of foreign varieties to be strongly affected by “Climatic conditions”. No comment was reported by the authors with a view to interpreting the yield and yield-related traits in the light of climatic conditions reported in table 14. The same consideration is applied to all traits.

Response: The correlation of the performance of the tested varieties with climatic conditions is given by what we have stated above. The CV was calculated according to the production values obtained in each of the test years and not according to the average production of the applied technological management.

Section 2.7. The discussion should be reported after the results in order to be interpreted in perspective of previous studies. Clarify better this section and check the English.

Response: In the revised form of the manuscript we have reported the discussions after the results and clarified this section better by adding some details as you suggested. We have also proofread the English throughout the manuscript.

Section 2.8.

Line 272. Please, report firstly the results and then discussions. As authors stated that ‘earlier studies showed that an increase in yield decreases protein content’, some references should be added.

Response: In the revised form of the manuscript we have reported the discussions after the results and clarified this section better by adding new references as you suggested.

Line 273. The result reported in reference 47 was limited to the two winter wheat varieties analyzed in that study, so the authors should add a clarification like ‘for genotype-specific’. Write better the period reported in lines 281-286, particularly the last sentence.

Response: Lines 281-286 reflect our findings and we consider what we have written to be in full agreement with them. Upon your suggestion we have modified the last sentence for better understanding: 'The current trend of climate change has rendered the notion of "normal" as relative'.

Section 2.9. Check the expression ‘culture system’. No discussion was reported for this trait.

Response: Thank you for your input. We have changed the expression "culture system" to "technological management". We have also introduced discussions.

Section 2.10. Check English and add discussion.

Response: We have proofread the English throughout the manuscript.

In the revised manuscript we have added discussions.

Section 2.11. No discussion. English required.

Response: In the revised manuscript we have added discussions. We have also proofread the English throughout the manuscript.

Section 2.12. No discussion. English required.

Response: In the revised manuscript we have added discussions.

Point 7. As in Table 13 is reported a summary of different management technologies applied without a comment, the same could have been moved to the supplementary materials.

Tables should be reviewed, avoiding orthographic mistakes and uniforming the significance (sometimes reported with asterisk and sometimes with zero number).

Response 7: Table 13 is a summary of all characters studied. We considered that an overview reflected in this synthetic form is easier to remember and to consult.

As much as we appreciate your comments, the meanings from the tables cannot be standardised because they reflect different things and it is up to the interpretation of the results.

Interpretation of results obtained using SP2F is very often used by researchers. Thus, reporting to the control can have:

  1. a) Positive results (increases) which are noted with stars as follows:

value between LD 5% and LD 1% - is marked with * and represents a significant increase;

value between LD 1% and LD 0.1% - is marked with ** and represents a distinctly significant increase;

value above LD 0.1% - is marked with *** and represents a very significant increase;

b)Negative results (decreases) which are marked with circles as follows:

value lying between LD 5% and LD 1% - is marked with a and represents a significant decrease;

value between LD 1% and LD 0.1% - is marked with oo and represents a distinctly significant decrease;

value above LD 0.1% - is marked with ooo and represents a very significant decrease.

Point 8. Materials and methods

In Table 14, no multiannual average was reported for relative humidity. Please, substitute Roman numbers with months.

Response 8: There is no data available on the multi-year average humidity. Multi-year averages are calculated over 60 years. Following your suggestion, we have made the appropriate changes by replacing Roman numbers with months.

Point 9. Conclusions

The conclusions are confusing and should be reviewed, in particular highlighting that productivity and proteins are positively correlated under certain conditions and that this requires further in-depth studies

Check the expression: very significant differences line 466; statistically inferior line 473, very significantly higher lines 482-483.

Response 9: In the revised manuscript we have improved the Conclusions section according to your suggestions, pointing out that productivity and proteins are positively correlated under certain conditions and that this requires further in-depth studies.

Regarding the expressions related to meanings, we answered above (Response 7).

Thank you for your time and valuable suggestions.

Yours faithfully,

The Authors

Response to Reviewer 1 Comments

Dear Reviewer,

First of all, we thank you for accepting the revision of our manuscript as well as for all the comments and suggestions provided.

Point 1. The paper entitled ‘Results Regarding the Influence of Some Different Cropping Systems on the Yield, Quality, Productivity Elements and Morphological Characters in Wheat (Triticum aestivum)’ (Plants-2496703)’ regards an overview on the effect of different cropping systems on morphology, yield and quality. A lack of discussion was evidenced, also in the light of climatic conditions. A revision of English is required. Also, Tables needed a revision for various mistakes and improve making them more presentable.

Although the topic is of interest, it cannot be accepted in this form but needed major revisions.

Response 1: In the revised version of the manuscript we have considered each of these suggestions. We have proofread the English throughout the manuscript.

Point 2. Title

The title should be revised as follows “The effect of different cropping systems on yield, quality, productivity elements and morphological characters in wheat (Triticum aestivum)”

Response 2: Thank you for the suggestion. The title has been changed accordingly.

Point 3. Abstract

According to the instructions for authors ‘The abstract should be a single paragraph and should follow the style of structured abstracts, but without headings: 1) Background: Place the question addressed in a broad context and highlight the purpose of the study; 2) Methods: Describe briefly the main methods or treatments applied. Include any relevant preregistration numbers, and species and strains of any animals used; 3) Results: Summarize the article's main findings; and 4) Conclusion: Indicate the main conclusions or interpretations. The abstract should be an objective representation of the article: it must not contain results which are not presented and substantiated in the main text and should not exaggerate the main conclusions’. The abstract here reported is unbalanced as it contains too many details of the experimental site and treatments that should be in the Materials and methods section and few results. Please, check carefully.

Response 3: Thank you for your suggestions. We have revised the Abstract section accordingly so that it now more clearly summarizes the central idea of the whole manuscript. We have removed details related to the experimental site and added some conclusive results.

Point 4. Keywords

Check the keywords, in particular ‘traits’. It should be better to introduce morphology and change management in technological management.

Response 4: We have modified the keywords as per your suggestion.

Point 5. Introduction

Point 5.1. Line 57. What is the logical sequence between the ‘security’ of line 57 with the ‘calories’ of the following lines? Moreover, wheat is one of the three main cereals. Which are the others?

Response 5.1: There are two distinct sentences here, in which we wanted, on the one hand, to stress the importance of wheat in ensuring food security, given that wheat is one of the most widely grown cereals, alongside rice and corn (we have now added this in the revised manuscript). On the other hand, we wanted to emphasise that since most wheat production is consumed in human nutrition, consumers supplement 60% of their daily calorie needs with wheat-based foods.

Point 5.2. Line 79. It is not clear the difference between the ‘farmer practice’ respect to the ‘high-input’ approach.

Response 5.2: We have taken into account your observation and defined the two terms in the revised form of the manuscript.

Point 5.3. Line 104. Clarify better the expression ‘subsistence farming’ in the aim of the work.  Moreover, move on ‘productivity elements’ after yield.

Response 5.3: Following your suggestion, we have clarified the term 'subsistence farming' in the paragraph regarding the purpose of our paper, giving some details to better express the essence of this type of agriculture. We have also moved 'productivity elements' after 'yield', as you have suggested, thank you.

Point 6. Results and Discussion

Section 2.1. Although this section is within the Results and Discussion one, the authors reported mainly the results without proper discussion and references to the literature in the field. Please, improve this section.

Response: Thank you for your comment. In the revised manuscript we have improved this section.

Section 2.2. Lines 123-126. The discussion should be reported after the results in order to be interpreted in perspective of previous studies. Moreover, the discussion reported in the lines 123-126 refers to the coefficient of variation as a good criterion for height stability, but this trait has not been considered by the authors in the manuscript. Review carefully.

Response: We have reported discussions immediately after the results, as you suggested, thank you.

Regarding the coefficient of variability, it is a simple bibliographical reference that wants to emphasize the role of plant size rather than yield as criteria for assessing stability. We wanted to show why we took into account the presentation of this characteristic and did not intend to present the coefficient of variability of the size. But in order not to create any confusion, we have improved the paper in this respect with a new table (Table 6) and a new figure (Figure 2).

Section 2.3. Clarify better the expression ‘semi-significant’ difference. Also, uniform the significance among the Tables (sometimes expressed with number and others with asterisk). The discussion reported in lines 154-155 requires to be implemented.

Response: That is correct, thank you for your observation. There is no such term as semi-significant it is only significant. It was an error that we fixed.

For tables, the numbers represent a difference which is a quantitative expression and the asterisk or circle represents a significance based on the fit within the LD values. Thus,

- the asterisk represents an increase: * -significant

                                                         ** - distinctly significant

                                                         ***-very significant

- the circle represents a decrease: o -significant

                                                     oo - distinctly significant

                                                     ooo-very significant

The discussion reported at lines 154-155 has been implemented as you suggested.

Section 2.4. No discussion has been reported.

Response: Thank you for your remark. In the revised manuscript we have added some discussion within this section.

Section 2.5. The text reported in lines 174-178 is to be revisited. Also check lines 184-207, sometimes confusing.

Response: We considered that if we tested 25 varieties we should also show some data on their yield and stability. Thus, we shifted from presenting the influence of the applied technology on yield to highlighting the varieties according to the results obtained through the coefficient of variability.

Lines 222-229. Check the English language. As the authors in these lines and previously (see section 2.2) mentioned the coefficient of variation (CV), the authors should show data of CV in a Table. Moreover, the authors should link better the two periods reported in lines 222-225 and 226-229 in order to define the best stable cultivars in terms of yield evidenced in lines 230-234.

Response: We have proofread the English throughout the manuscript.

We did not present in detail in order not to overload the paper, showing only what was essential. The first CV, as specified within the text, is calculated according to the management applied (between the 4 variants - low-input conventional, conventional, delayed conventional and subsistence). We have introduced a new figure in this regard for a better understanding of the text (Figure 2).

The second CV is calculated according to the yields obtained in each of the years of the experiment, regardless of the technology applied. We have introduced a new table for better understanding of the text (Table 6).

For linking better discussions on CV, as you recommended, we have removed this paragraph:

The coefficients of determination (r2) highlighted the fact that 67% of the variability in conventional yield is matched by variability in conventional low-input yield. Also, 57% of the variability in conventional yield is matched by variability in subsistence system yield but only 41% by variability in conventional delayed yield”.

Lines 231- 232. The authors justify the different performance of foreign varieties to be strongly affected by “Climatic conditions”. No comment was reported by the authors with a view to interpreting the yield and yield-related traits in the light of climatic conditions reported in table 14. The same consideration is applied to all traits.

Response: The correlation of the performance of the tested varieties with climatic conditions is given by what we have stated above. The CV was calculated according to the production values obtained in each of the test years and not according to the average production of the applied technological management.

Section 2.7. The discussion should be reported after the results in order to be interpreted in perspective of previous studies. Clarify better this section and check the English.

Response: In the revised form of the manuscript we have reported the discussions after the results and clarified this section better by adding some details as you suggested. We have also proofread the English throughout the manuscript.

Section 2.8.

Line 272. Please, report firstly the results and then discussions. As authors stated that ‘earlier studies showed that an increase in yield decreases protein content’, some references should be added.

Response: In the revised form of the manuscript we have reported the discussions after the results and clarified this section better by adding new references as you suggested.

Line 273. The result reported in reference 47 was limited to the two winter wheat varieties analyzed in that study, so the authors should add a clarification like ‘for genotype-specific’. Write better the period reported in lines 281-286, particularly the last sentence.

Response: Lines 281-286 reflect our findings and we consider what we have written to be in full agreement with them. Upon your suggestion we have modified the last sentence for better understanding: 'The current trend of climate change has rendered the notion of "normal" as relative'.

Section 2.9. Check the expression ‘culture system’. No discussion was reported for this trait.

Response: Thank you for your input. We have changed the expression "culture system" to "technological management". We have also introduced discussions.

Section 2.10. Check English and add discussion.

Response: We have proofread the English throughout the manuscript.

In the revised manuscript we have added discussions.

Section 2.11. No discussion. English required.

Response: In the revised manuscript we have added discussions. We have also proofread the English throughout the manuscript.

Section 2.12. No discussion. English required.

Response: In the revised manuscript we have added discussions.

Point 7. As in Table 13 is reported a summary of different management technologies applied without a comment, the same could have been moved to the supplementary materials.

Tables should be reviewed, avoiding orthographic mistakes and uniforming the significance (sometimes reported with asterisk and sometimes with zero number).

Response 7: Table 13 is a summary of all characters studied. We considered that an overview reflected in this synthetic form is easier to remember and to consult.

As much as we appreciate your comments, the meanings from the tables cannot be standardised because they reflect different things and it is up to the interpretation of the results.

Interpretation of results obtained using SP2F is very often used by researchers. Thus, reporting to the control can have:

  1. a) Positive results (increases) which are noted with stars as follows:

value between LD 5% and LD 1% - is marked with * and represents a significant increase;

value between LD 1% and LD 0.1% - is marked with ** and represents a distinctly significant increase;

value above LD 0.1% - is marked with *** and represents a very significant increase;

b)Negative results (decreases) which are marked with circles as follows:

value lying between LD 5% and LD 1% - is marked with a and represents a significant decrease;

value between LD 1% and LD 0.1% - is marked with oo and represents a distinctly significant decrease;

value above LD 0.1% - is marked with ooo and represents a very significant decrease.

Point 8. Materials and methods

In Table 14, no multiannual average was reported for relative humidity. Please, substitute Roman numbers with months.

Response 8: There is no data available on the multi-year average humidity. Multi-year averages are calculated over 60 years. Following your suggestion, we have made the appropriate changes by replacing Roman numbers with months.

Point 9. Conclusions

The conclusions are confusing and should be reviewed, in particular highlighting that productivity and proteins are positively correlated under certain conditions and that this requires further in-depth studies

Check the expression: very significant differences line 466; statistically inferior line 473, very significantly higher lines 482-483.

Response 9: In the revised manuscript we have improved the Conclusions section according to your suggestions, pointing out that productivity and proteins are positively correlated under certain conditions and that this requires further in-depth studies.

Regarding the expressions related to meanings, we answered above (Response 7).

Thank you for your time and valuable suggestions.

Yours faithfully,

The Authors

Response to Reviewer 1 Comments

Dear Reviewer,

First of all, we thank you for accepting the revision of our manuscript as well as for all the comments and suggestions provided.

Point 1. The paper entitled ‘Results Regarding the Influence of Some Different Cropping Systems on the Yield, Quality, Productivity Elements and Morphological Characters in Wheat (Triticum aestivum)’ (Plants-2496703)’ regards an overview on the effect of different cropping systems on morphology, yield and quality. A lack of discussion was evidenced, also in the light of climatic conditions. A revision of English is required. Also, Tables needed a revision for various mistakes and improve making them more presentable.

Although the topic is of interest, it cannot be accepted in this form but needed major revisions.

Response 1: In the revised version of the manuscript we have considered each of these suggestions. We have proofread the English throughout the manuscript.

Point 2. Title

The title should be revised as follows “The effect of different cropping systems on yield, quality, productivity elements and morphological characters in wheat (Triticum aestivum)”

Response 2: Thank you for the suggestion. The title has been changed accordingly.

Point 3. Abstract

According to the instructions for authors ‘The abstract should be a single paragraph and should follow the style of structured abstracts, but without headings: 1) Background: Place the question addressed in a broad context and highlight the purpose of the study; 2) Methods: Describe briefly the main methods or treatments applied. Include any relevant preregistration numbers, and species and strains of any animals used; 3) Results: Summarize the article's main findings; and 4) Conclusion: Indicate the main conclusions or interpretations. The abstract should be an objective representation of the article: it must not contain results which are not presented and substantiated in the main text and should not exaggerate the main conclusions’. The abstract here reported is unbalanced as it contains too many details of the experimental site and treatments that should be in the Materials and methods section and few results. Please, check carefully.

Response 3: Thank you for your suggestions. We have revised the Abstract section accordingly so that it now more clearly summarizes the central idea of the whole manuscript. We have removed details related to the experimental site and added some conclusive results.

Point 4. Keywords

Check the keywords, in particular ‘traits’. It should be better to introduce morphology and change management in technological management.

Response 4: We have modified the keywords as per your suggestion.

Point 5. Introduction

Point 5.1. Line 57. What is the logical sequence between the ‘security’ of line 57 with the ‘calories’ of the following lines? Moreover, wheat is one of the three main cereals. Which are the others?

Response 5.1: There are two distinct sentences here, in which we wanted, on the one hand, to stress the importance of wheat in ensuring food security, given that wheat is one of the most widely grown cereals, alongside rice and corn (we have now added this in the revised manuscript). On the other hand, we wanted to emphasise that since most wheat production is consumed in human nutrition, consumers supplement 60% of their daily calorie needs with wheat-based foods.

Point 5.2. Line 79. It is not clear the difference between the ‘farmer practice’ respect to the ‘high-input’ approach.

Response 5.2: We have taken into account your observation and defined the two terms in the revised form of the manuscript.

Point 5.3. Line 104. Clarify better the expression ‘subsistence farming’ in the aim of the work.  Moreover, move on ‘productivity elements’ after yield.

Response 5.3: Following your suggestion, we have clarified the term 'subsistence farming' in the paragraph regarding the purpose of our paper, giving some details to better express the essence of this type of agriculture. We have also moved 'productivity elements' after 'yield', as you have suggested, thank you.

Point 6. Results and Discussion

Section 2.1. Although this section is within the Results and Discussion one, the authors reported mainly the results without proper discussion and references to the literature in the field. Please, improve this section.

Response: Thank you for your comment. In the revised manuscript we have improved this section.

Section 2.2. Lines 123-126. The discussion should be reported after the results in order to be interpreted in perspective of previous studies. Moreover, the discussion reported in the lines 123-126 refers to the coefficient of variation as a good criterion for height stability, but this trait has not been considered by the authors in the manuscript. Review carefully.

Response: We have reported discussions immediately after the results, as you suggested, thank you.

Regarding the coefficient of variability, it is a simple bibliographical reference that wants to emphasize the role of plant size rather than yield as criteria for assessing stability. We wanted to show why we took into account the presentation of this characteristic and did not intend to present the coefficient of variability of the size. But in order not to create any confusion, we have improved the paper in this respect with a new table (Table 6) and a new figure (Figure 2).

Section 2.3. Clarify better the expression ‘semi-significant’ difference. Also, uniform the significance among the Tables (sometimes expressed with number and others with asterisk). The discussion reported in lines 154-155 requires to be implemented.

Response: That is correct, thank you for your observation. There is no such term as semi-significant it is only significant. It was an error that we fixed.

For tables, the numbers represent a difference which is a quantitative expression and the asterisk or circle represents a significance based on the fit within the LD values. Thus,

- the asterisk represents an increase: * -significant

                                                         ** - distinctly significant

                                                         ***-very significant

- the circle represents a decrease: o -significant

                                                     oo - distinctly significant

                                                     ooo-very significant

The discussion reported at lines 154-155 has been implemented as you suggested.

Section 2.4. No discussion has been reported.

Response: Thank you for your remark. In the revised manuscript we have added some discussion within this section.

Section 2.5. The text reported in lines 174-178 is to be revisited. Also check lines 184-207, sometimes confusing.

Response: We considered that if we tested 25 varieties we should also show some data on their yield and stability. Thus, we shifted from presenting the influence of the applied technology on yield to highlighting the varieties according to the results obtained through the coefficient of variability.

Lines 222-229. Check the English language. As the authors in these lines and previously (see section 2.2) mentioned the coefficient of variation (CV), the authors should show data of CV in a Table. Moreover, the authors should link better the two periods reported in lines 222-225 and 226-229 in order to define the best stable cultivars in terms of yield evidenced in lines 230-234.

Response: We have proofread the English throughout the manuscript.

We did not present in detail in order not to overload the paper, showing only what was essential. The first CV, as specified within the text, is calculated according to the management applied (between the 4 variants - low-input conventional, conventional, delayed conventional and subsistence). We have introduced a new figure in this regard for a better understanding of the text (Figure 2).

The second CV is calculated according to the yields obtained in each of the years of the experiment, regardless of the technology applied. We have introduced a new table for better understanding of the text (Table 6).

For linking better discussions on CV, as you recommended, we have removed this paragraph:

The coefficients of determination (r2) highlighted the fact that 67% of the variability in conventional yield is matched by variability in conventional low-input yield. Also, 57% of the variability in conventional yield is matched by variability in subsistence system yield but only 41% by variability in conventional delayed yield”.

Lines 231- 232. The authors justify the different performance of foreign varieties to be strongly affected by “Climatic conditions”. No comment was reported by the authors with a view to interpreting the yield and yield-related traits in the light of climatic conditions reported in table 14. The same consideration is applied to all traits.

Response: The correlation of the performance of the tested varieties with climatic conditions is given by what we have stated above. The CV was calculated according to the production values obtained in each of the test years and not according to the average production of the applied technological management.

Section 2.7. The discussion should be reported after the results in order to be interpreted in perspective of previous studies. Clarify better this section and check the English.

Response: In the revised form of the manuscript we have reported the discussions after the results and clarified this section better by adding some details as you suggested. We have also proofread the English throughout the manuscript.

Section 2.8.

Line 272. Please, report firstly the results and then discussions. As authors stated that ‘earlier studies showed that an increase in yield decreases protein content’, some references should be added.

Response: In the revised form of the manuscript we have reported the discussions after the results and clarified this section better by adding new references as you suggested.

Line 273. The result reported in reference 47 was limited to the two winter wheat varieties analyzed in that study, so the authors should add a clarification like ‘for genotype-specific’. Write better the period reported in lines 281-286, particularly the last sentence.

Response: Lines 281-286 reflect our findings and we consider what we have written to be in full agreement with them. Upon your suggestion we have modified the last sentence for better understanding: 'The current trend of climate change has rendered the notion of "normal" as relative'.

Section 2.9. Check the expression ‘culture system’. No discussion was reported for this trait.

Response: Thank you for your input. We have changed the expression "culture system" to "technological management". We have also introduced discussions.

Section 2.10. Check English and add discussion.

Response: We have proofread the English throughout the manuscript.

In the revised manuscript we have added discussions.

Section 2.11. No discussion. English required.

Response: In the revised manuscript we have added discussions. We have also proofread the English throughout the manuscript.

Section 2.12. No discussion. English required.

Response: In the revised manuscript we have added discussions.

Point 7. As in Table 13 is reported a summary of different management technologies applied without a comment, the same could have been moved to the supplementary materials.

Tables should be reviewed, avoiding orthographic mistakes and uniforming the significance (sometimes reported with asterisk and sometimes with zero number).

Response 7: Table 13 is a summary of all characters studied. We considered that an overview reflected in this synthetic form is easier to remember and to consult.

As much as we appreciate your comments, the meanings from the tables cannot be standardised because they reflect different things and it is up to the interpretation of the results.

Interpretation of results obtained using SP2F is very often used by researchers. Thus, reporting to the control can have:

  1. a) Positive results (increases) which are noted with stars as follows:

value between LD 5% and LD 1% - is marked with * and represents a significant increase;

value between LD 1% and LD 0.1% - is marked with ** and represents a distinctly significant increase;

value above LD 0.1% - is marked with *** and represents a very significant increase;

b)Negative results (decreases) which are marked with circles as follows:

value lying between LD 5% and LD 1% - is marked with a and represents a significant decrease;

value between LD 1% and LD 0.1% - is marked with oo and represents a distinctly significant decrease;

value above LD 0.1% - is marked with ooo and represents a very significant decrease.

Point 8. Materials and methods

In Table 14, no multiannual average was reported for relative humidity. Please, substitute Roman numbers with months.

Response 8: There is no data available on the multi-year average humidity. Multi-year averages are calculated over 60 years. Following your suggestion, we have made the appropriate changes by replacing Roman numbers with months.

Point 9. Conclusions

The conclusions are confusing and should be reviewed, in particular highlighting that productivity and proteins are positively correlated under certain conditions and that this requires further in-depth studies

Check the expression: very significant differences line 466; statistically inferior line 473, very significantly higher lines 482-483.

Response 9: In the revised manuscript we have improved the Conclusions section according to your suggestions, pointing out that productivity and proteins are positively correlated under certain conditions and that this requires further in-depth studies.

Regarding the expressions related to meanings, we answered above (Response 7).

Thank you for your time and valuable suggestions.

Yours faithfully,

The Authors

Reviewer 2 Report

The manuscript titled “Results regarding the influence of some different cropping systems on the yield, quality, productivity elements and morphological characters in wheat (Triticum aestivum)” presents a study on the impact of various cropping systems on increasing grain yield and quality. wheat. Winter wheat is a strategic grain crop, the grain of which is used for food and fodder purposes. The study is of interest from the point of view of increasing the productivity of crops and solving food security for many countries of the world.

However, there are a number of questions for the study.

1. Which varieties were included in the study, hard or soft? Usually soft varieties give a higher yield than hard ones.

2. What system was used to fertilize? Were they applied once in autumn or spring, or differentially? So, macroelements, when they are introduced into the soil, have different mobility along the horizons. Nitrogen has a high mobility in the soil, so it makes no sense to introduce it in winter. After sowing, the crop will be able to use only a small amount of this nutrient. Accordingly, its effect and influence on cultures will not be significant.

3. Did you take into account the removal of nutrients with stubble? How were the optimal doses of fertilizers calculated?

Author Response

Response to Reviewer 2 Comments

Dear Reviewer,

First of all, we thank you for accepting the revision of our manuscript as well as for all the comments and suggestions provided.

The manuscript titled “Results regarding the influence of some different cropping systems on the yield, quality, productivity elements and morphological characters in wheat (Triticum aestivum)” presents a study on the impact of various cropping systems on increasing grain yield and quality wheat. Winter wheat is a strategic grain crop, the grain of which is used for food and fodder purposes. The study is of interest from the point of view of increasing the productivity of crops and solving food security for many countries of the world.

However, there are a number of questions for the study.

Point 1. Which varieties were included in the study, hard or soft? Usually soft varieties give a higher yield than hard ones.

Response 1: The varieties included in the study are soft varieties.

Point 2. What system was used to fertilize? Were they applied once in autumn or spring, or differentially? So, macroelements, when they are introduced into the soil, have different mobility along the horizons. Nitrogen has a high mobility in the soil, so it makes no sense to introduce it in winter. After sowing, the crop will be able to use only a small amount of this nutrient. Accordingly, its effect and influence on cultures will not be significant.

Response 2: Factor B - fertilization level had 2 gradations: N40P40 and N100P40, in 3 replications. The first level of fertilization corresponds to the conventional low-input system and the second to the conventional system.

N40P40 fertilization level was ensured by applying 200 kg/ha NPK 20:20:0 complex fertilizer in autumn, incorporated into the soil before sowing. No fertilizer was applied in spring.

N100P40 fertilization level was ensured by applying 200 kg/ha NPK 20:20:0 complex fertilizer in autumn, incorporated into the soil before sowing, and NH4NO3 at 34.5% s.a., applied in spring. This type of fertilization is the most common in Romanian farms.

We, as researchers, recommend farmers fertilizing in autumn with complex fertilizers with a lower nitrogen and higher phosphorus content, (e.g. type 18 :46 :0), which ensure a good start for the wheat crop during vegetation, good tillering and increase the plants' resistance to frost. In the present study, as far as fertilization is concerned, we considered it necessary to take into account the practice used in production farms.

Point 3. Did you take into account the removal of nutrients with stubble? How were the optimal doses of fertilizers calculated?

Response 3: Removal of nutrients with stubble was not taken into account. Optimal doses should be calculated according to the macro and micro nutrient content of the soil but, as stated above, we have taken into account the practice used at farms. The segment of those who calculate their doses according to the macro nutrient content in the soil is extremely low and therefore the cost of analysis is high. The aim of our paper was to persuade subsistence farmers (over 600,000 subsistence farmers who own about 50% of the agricultural area, achieve only 10-15% of the agricultural production) to at least align themselves to the level of farmers applying the low-input conventional and/or conventional system.

Romania's agricultural sector has a high proportion of small farms below 5 ha - estimated at over 93% of the country's agricultural population. This is by far the highest share of subsistence and semi-subsistence farms in the EU, producing goods mainly for own consumption.

Thank you for your time and valuable suggestions.

Yours faithfully,

The Authors

Round 2

Reviewer 1 Report

The authors have made a great effort to improve the manuscript, but some revisions are still needed before accepting the article.

Abstract Line 18: substitute ‘range’ with ‘collection’.

Line 20: delete ‘production’ before quality.

Lines 28-29. Improve this sentence ‘It should be noted that obtaining a quality yield is based on nitrogen supply’. An example is reported here ‘Nitrogen supply is the most important factor for determining wheat productivity and grain quality’.

Lines 166-170. The meaning of farmer practice is too long in that context. Please, summarize in a few words the farmer practice and high-input agriculture.

Lines 199-205. Summarize the concept, remembering that this the aim of the scientific article. So, the authors should avoid the comments in lines 202-205 or just few important words.

The response to the section 2.3 regarding the asterisk, the circle etc should be reported, in a concise way, in the legend of the Tables. Please check.

Lines 222-224, section 2.1, please improve the meaning of the sentence.

Line 292. Change ‘determined’ with ‘influenced’.

Lines 300-305. It is uncorrected to use this expression ‘In 2008-2009 crop year an experience took place’. The authors have to re-write this period in a more professional form, with the reference at the end. It is not a   report but a scientific paper.

Line 306. The same for the expression ‘It is expected that by 2050, the global demand for food will increase by 70% [46]’. It is not a report. This sentence could be used for example in the conclusions if the authors would add a solution to the increasing demand of food.

Lines 387-393. This period is more appropriate as a conclusion rather than a discussion. Please, try to make it more suitable for this section or move it to the general conclusions.

Lines 445-452. Please, check these lines in order to be more appropriate for a scientific article rather than for a report.

Lines 464-465. Please, clarify the sentence ‘The current trend of climate change has rendered the notion of "normal" as relative’.

Lines 487-499. Please, write in a more appropriate form, deleting for example expressions as ‘through their results’, etc. Firstly the authors reported a ‘simultaneous increase yield and protein content’ than they discuss about ‘the lower grain protein concentration..’. Please, re-write in a more logical way up to a hypothesis.

Lines 511-520. Please, write discussion in a logically mode, comparing your data on a specific trait as Wet gluten content with literature, not gluten index specific of paragraph 2.12.

Lines 530-531. Sedimentation and Zeleny test are not the same? Please, check.

Lines 583-591. Please rearrange the text as a legend.

Conclusions might be improved as resulted similar to the abstract. Also an English revision is required.

Minor revision of English is needed.

Author Response

Response to Reviewer 1 Comments

Dear Reviewer,

First of all, we thank you for accepting the revision of our manuscript as well as for all the comments and suggestions provided.

Point 1. Abstract Line 18: substitute ‘range’ with ‘collection’.

Response 1: As per your suggestion, we have replaced ‘range’ with ‘collection’.

Point 2. Line 20: delete ‘production’ before quality.

Response 2: We have deleted ‘production’.

Point 3. Lines 28-29. Improve this sentence ‘It should be noted that obtaining a quality yield is based on nitrogen supply’. An example is reported here ‘Nitrogen supply is the most important factor for determining wheat productivity and grain quality’.

Response 3: We have replaced that sentence with the one you suggested.

Point 4. Lines 166-170. The meaning of farmer practice is too long in that context. Please, summarize in a few words the farmer practice and high-input agriculture.

Response 4: We have modified it accordingly.

Point 5. Lines 199-205. Summarize the concept, remembering that this the aim of the scientific article. So, the authors should avoid the comments in lines 202-205 or just few important words.

Response  5: We have amended that paragraph accordingly.

Point 6. The response to the section 2.3 regarding the asterisk, the circle etc should be reported, in a concise way, in the legend of the Tables. Please check.

Response 6: As recommended, we have inserted after each table the explanation of the meanings.

Point 7. Lines 222-224, section 2.1, please improve the meaning of the sentence.

Response 7: We have modified it accordingly.

Point 8. Line 292. Change ‘determined’ with ‘influenced’.

Response 8: As per your suggestion, we have replaced ‘determined’ with ‘influenced’.

Point 9. Lines 300-305. It is uncorrected to use this expression ‘In 2008-2009 crop year an experience took place’. The authors have to re-write this period in a more professional form, with the reference at the end. It is not a   report but a scientific paper.

Response 9: We have amended that paragraph accordingly.

Point 10. Line 306. The same for the expression ‘It is expected that by 2050, the global demand for food will increase by 70% [46]’. It is not a report. This sentence could be used for example in the conclusions if the authors would add a solution to the increasing demand of food.

Response 10: We have deleted that sentence.

Point 11. Lines 387-393. This period is more appropriate as a conclusion rather than a discussion. Please, try to make it more suitable for this section or move it to the general conclusions.

Response 11: We have modified that paragraph accordingly.

Point 12. Lines 445-452. Please, check these lines in order to be more appropriate for a scientific article rather than for a report.

Response 12: We have eliminated the details.

Point 13. Lines 464-465. Please, clarify the sentence ‘The current trend of climate change has rendered the notion of "normal" as relative’.

Response 13: We have deleted the sentence to avoid any confusion.

Point 14. Lines 487-499. Please, write in a more appropriate form, deleting for example expressions as ‘through their results’, etc. Firstly the authors reported a ‘simultaneous increase yield and protein content’ than they discuss about ‘the lower grain protein concentration..’. Please, re-write in a more logical way up to a hypothesis.

Response 14: For a better understanding of the text we have rephrased that paragraph.

Point 15. Lines 511-520. Please, write discussion in a logically mode, comparing your data on a specific trait as Wet gluten content with literature, not gluten index specific of paragraph 2.12.

Response 15: We have now deleted the term 'gluten index', since it is specific to paragraph 2.12.  

Point 16. Lines 530-531. Sedimentation and Zeleny test are not the same? Please, check.

Response 16: Thank you for your comment. We have modified it accordingly.

Point 17. Lines 583-591. Please rearrange the text as a legend.

Response 17: We have modified it accordingly.

Point 18. Conclusions might be improved as resulted similar to the abstract. Also an English revision is required.

Response 18: We have improved this section. Also, the entire manuscript has been proof-read by a certified English translator.

Thank you for your time and valuable suggestions.

Yours faithfully,

The Authors
